# *Bmal1* function in skeletal muscle regulates sleep

J Christopher Ehlen[1†], Allison J Brager[1,2†], Julie Baggs[1], Lennisha Pinckney[1], Cloe L Gray[1], Jason P DeBruyne[1], Karyn A Esser[3], Joseph S Takahashi[4], Ketema N Paul[1,5*]

[1]Neuroscience Institute, Morehouse School of Medicine, Atlanta, United States; [2]Behavioral Biology Branch, Center for Military Psychiatry and Neuroscience, Walter Reed Army Institute of Research, Silver Spring, United States; [3]Myology Institute, College of Medicine, University of Florida, Gainesville, United States; [4]Department of Neuroscience, Howard Hughes Medical Institute, University of Texas Southwestern Medical Center, Dallas, United States; [5]Department of Integrative Biology and Physiology, University of California, Los Angeles, California, United States

**Abstract** Sleep loss can severely impair the ability to perform, yet the ability to recover from sleep loss is not well understood. Sleep regulatory processes are assumed to lie exclusively within the brain mainly due to the strong behavioral manifestations of sleep. Whole-body knockout of the circadian clock gene *Bmal1* in mice affects several aspects of sleep, however, the cells/tissues responsible are unknown. We found that restoring *Bmal1* expression in the brains of *Bmal1*-knockout mice did not rescue *Bmal1*-dependent sleep phenotypes. Surprisingly, most sleep-amount, but not sleep-timing, phenotypes could be reproduced or rescued by knocking out or restoring BMAL1 exclusively in skeletal muscle, respectively. We also found that overexpression of skeletal-muscle *Bmal1* reduced the recovery response to sleep loss. Together, these findings demonstrate that *Bmal1* expression in skeletal muscle is both necessary and sufficient to regulate total sleep amount and reveal that critical components of normal sleep regulation occur in muscle.

*For correspondence: ketema.
paul@ucla.edu

†These authors contributed
equally to this work

Competing interest: See
page 12

Reviewing editor: Louis J
Ptáček, University of California,
San Francisco, United States

## Introduction

The ability to recover from sleep loss is critical for preserving cognitive processes and executive functioning (*McCoy and Strecker, 2011*; *Simon et al., 2015*; *Tucker et al., 2010*). The mechanisms that are responsible for the recovery from sleep loss are not well understood. Genetic deletion of the circadian transcription factor *Bmal1* (brain and muscle ARNT-like factor; gene symbol *Arntl*) in mice completely ablates circadian clock function (*Bunger et al., 2000*) and has effects on sleep that include: increased total sleep amount, increased non-rapid eye movement (NREM) sleep intensity and reduced ability to recover from sleep loss (*Laposky et al., 2005*). Because *Bmal1* whole-body deletion causes a broad range of physical abnormalities (e.g. reduced locomotor activity, joint abnormalities, reduced lifespan; *Bunger et al., 2005*; *Kondratov et al., 2006*), we sought to isolate sleep phenotypes from the potential effects of other phenotypes using tissue-specific *Bmal1* rescue and knockout models.

We first attempted to rescue electroencephalographic (EEG) sleep-phenotypes in *Bmal1* knockout mice by restoring functional *Bmal1* expression selectively in brain. To do this, we used a transgenic model that rescues *Bmal1* expression in the brain of *Bmal1* whole-body KO's, thus restoring circadian behavior (*Scg2::tTa; tetO::Bmal1-HA*; *McDearmon et al., 2006*). Surprisingly, transgenic

**eLife digest** We spend nearly one third of our lives asleep. Sleep plays a critical role in human health and is regulated by multiple brain regions. Genes are some of the factors that control sleep. Recent studies have shown that mice in which a gene called *Bmal1* had been completely removed, sleep more than mice that still have the gene. These *Bmal1*-deficient mice also respond differently to sleep loss. However, until now, it was not known which tissues and cells that carry active (or 'expressed') *Bmal1* are involved in regulating sleep.

To find out if *Bmal1* activity in the brain is sufficient to recover from sleep loss, Ehlen, Brager et al. compared genetically modified mice that either expressed *Bmal1* only in the brain, or only in the muscle tissue that covers the skeleton. After the mice were kept awake for six hours, their sleep was monitored by measuring electrical signals on the surface of the skull. Contrary to what they expected, Ehlen et al. found that mice with *Bmal1* expressed in the skeletal muscle were able to have a normal sleep pattern, while mice with *Bmal1* expressed in the brain had an abnormal sleep pattern.

Further experiments show that removing *Bmal1* from the skeletal muscle of mice, but allowing the gene to be expressed in other tissues, produced sleeping patterns that were similar to those seen in mice that were completely missing the *Bmal1* gene. These results indicate that *Bmal1* in skeletal muscle is important to help regulate sleep, and that the signal for sleepiness does not only originate from the brain.

This is the first study to show that skeletal muscle can regulate sleep. The next step will be to identify the specific signal the muscle uses to trigger the brain to sleep. Understanding the mechanisms that regulate sleep may help to develop new treatments for sleep disorders.

*Bmal1* expression in the brain (i.e., brain rescued) did not restore NREM sleep-amount—one of the most prominent sleep changes in whole-body *Bmal1* knockouts (*Figure 1A,B*). Since circadian rhythms of locomotor activity are restored in this model (*McDearmon et al., 2006*), our finding suggests that the sleep disturbances in *Bmal1* knockout mice are not exclusively the result of disrupted circadian behavior. More importantly, the results show that *Bmal1* expression in brain does not restore normal sleep amounts in *Bmal1* knockout mice, suggesting that expression in other tissues may be important.

To begin this investigation of other tissues, we chose mice harboring a transgene that restores *Bmal1* specifically in skeletal muscle, but does not restore circadian behavior (i.e., muscle rescued; *Acta1::Bmal1-HA*). This model rescues several muscle-related phenotypes despite non-cycling BMAL1 levels (*McDearmon et al., 2006*). To our surprise, although REM sleep was unaffected (*Figure 1B and C*), we found that restoring *Bmal1* in skeletal muscle completely restored NREM sleep amount to wild-type levels in otherwise *Bmal1*-deficient mice (*Figure 1C*). We also assessed whether *Bmal1* function in the skeletal muscle altered sleep intensity by measuring NREM slow wave activity (SWA, 0.5–4 Hz; *Borbély et al., 1981*; *Dijk et al., 1990*). In our hands, NREM SWA was not altered in *Bmal1* knockout mice; similarly, NREM SWA was not significantly altered by rescuing BMAL1 in brain or muscle (*Figure 1B and C*). These experiments demonstrate that restoring *Bmal1* in the skeletal muscle of otherwise *Bmal1*-deficient mice is sufficient to restore normal NREM sleep amount, independently of *Bmal1* expression in the brain. The diurnal rhythm in sleep amount, however, is not restored (*Figure 1A*).

To investigate the effect of *Bmal1* function on recovery from sleep loss, we subjected the *Bmal1* mouse lines to 6 hr of forced wakefulness and monitored sleep EEG's during recovery (*Figure 2A*). The amount and type of recovery sleep observed in response to forced wakefulness is commonly used to assess changes in sleep homeostasis (i.e., sleep drive)—this recovery is typically characterized by increased sleep amount and increased SWA (*Ehlen et al., 2013*). Whole-body KO of *Bmal1* did not affect NREM recovery-sleep amount (*Figure 2A,B*), but did significantly prevent increased NREM-SWA (*Figure 2C,D*). Rescue of *Bmal1* in either brain- or muscle-rescued mice reduced NREM recovery-sleep following forced wakefulness (*Figure 2A,B*), a finding that indicates *Bmal1* restoration in either tissue reduces sleep drive following forced wakefulness. Notably, NREM SWA in whole-

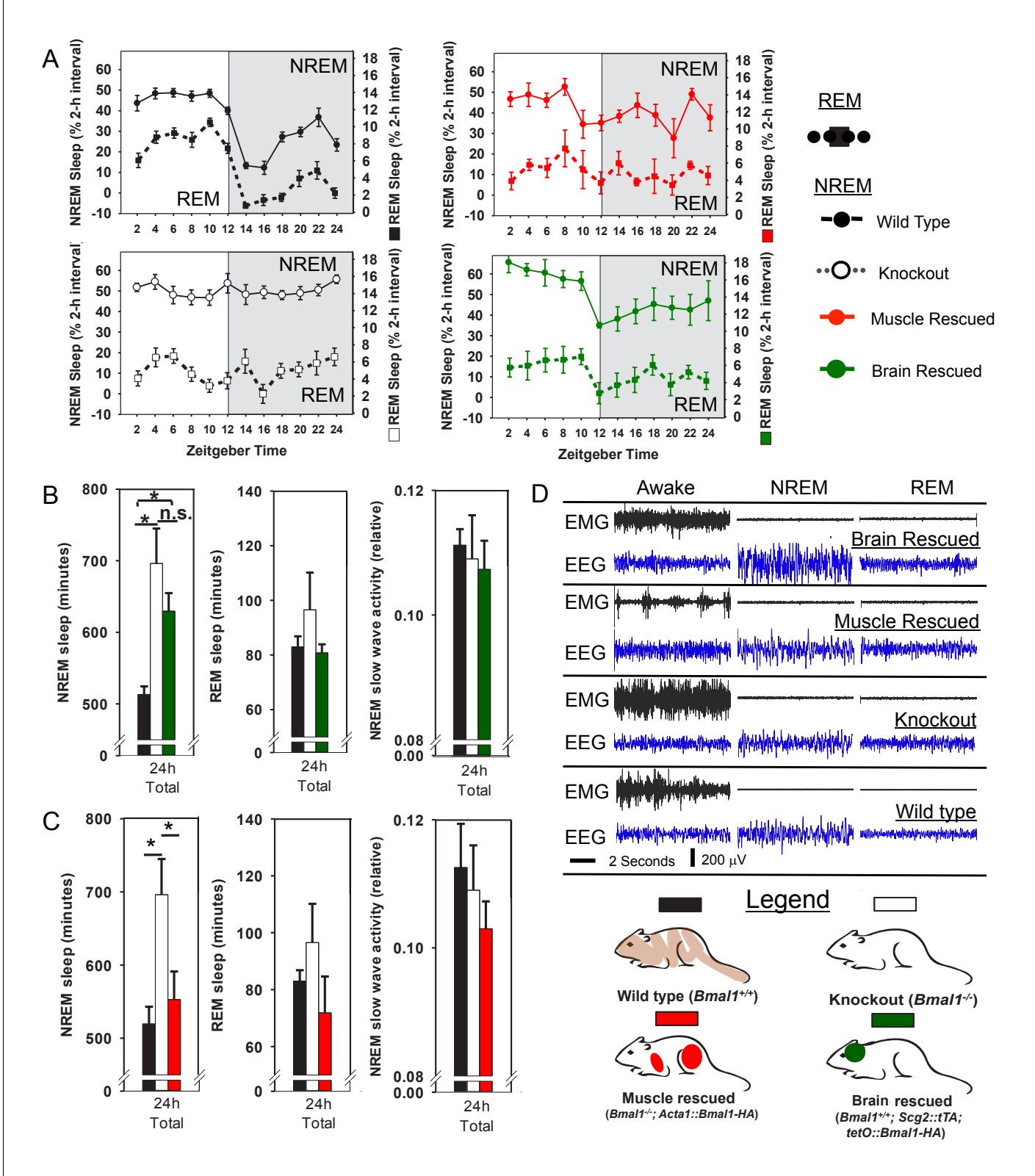

**Figure 1.** Rescuing *Bmal1* in skeletal muscle restores daily non-rapid eye movement (NREM) sleep amount. 24 hr electroencephalographic recordings were conducted in undisturbed mice listed in the legend. The 24 hr pattern of NREM and REM sleep are shown in A. Whole-body knockout of *Bmal1* significantly increased NREM sleep when compared to WT controls (B, ANOVA $F_{(2,27)}$=11.3, p=0.005; p<0.001, posthoc Tukey's test). Rescuing *Bmal1* in the brain of knockouts did not restore NREM sleep to WT levels (B, p=0.001 vs. WT; Tukey's test); however, the effect of *Bmal1* knockout was reversed

*Figure 1 continued on next page*

*Figure 1 continued*

when *Bmal1* was rescued in the skeletal muscle (C, ANOVA $F_{(2,31)}=9.9$, p<0.001; p=0.88 vs. WT, p<0.001 vs. KO, Tukey's test). No differences were found in REM sleep or NREM slow-wave activity. Knockout animals are replotted in B and C to aid in comparison. KO mice were offspring from independent crosses of heterozygous *Bmal1* KO's. Representative electroencephalographic recordings are displayed in D. Grey boxes indicate the active/dark period. Bars and points represent mean ± s.e.m. *, p<0.05. WT (brain) n = 11, knockout n = 12, muscle rescue n = 5, brain rescue n = 4, WT (muscle) n = 16.

The following figure supplement is available for figure 1:

**Figure supplement 1.** *Bmal1-HA* is not detectable in the Brains of *Acta1::Bmal1-HA* mice.

body KO mice was rescued by restoring *Bmal1* in skeletal muscle, but not brain (*Figure 2A,B*). These findings demonstrate that restoring *Bmal1* in the skeletal muscle of *Bmal1*-deficient mice is sufficient to restore normal SWA following sleep loss.

Our results indicate that the rescue of *Bmal1* in skeletal muscle is sufficient to restore both NREM sleep amount and the SWA recovery-responses to lost sleep. We next sought to determine if *Bmal1* in skeletal muscle was necessary for the effects we observed on sleep processes by specifically deleting *Bmal1* in skeletal muscle (*McCarthy et al., 2012a*). Mice lacking BMAL1 in their skeletal muscle had significantly increased baseline NREM sleep amount, a result similar to whole-body *Bmal1* knockout mice (*Figure 3A–C*). Moreover, when muscle-specific knockouts were examined after 6 hr of forced wakefulness, recovery sleep was nearly half that of WT mice (*Figure 3D*). Forced wakefulness also increased SWA in these mice when compared to controls (*Figure 3D*). This effect on SWA suggests that these mice have higher sleep intensity than WTs during recovery from sleep loss. Together, these data support a role for skeletal muscle and *Bmal1* in regulating the ability to recover from sleep loss. Moreover, these data support the conclusion that *Bmal1* expression in skeletal muscle is both necessary and sufficient for the regulation of normal NREM sleep amount.

That many of the sleep phenotypes caused by whole-body BMAL1 deficiency are either rescued by muscle-specific *Bmal1* expression or recapitulated by muscle-specific *Bmal1* deletion suggest that sleep phenotypes in the *Bmal1* knockout mice are, in whole or in part, due to loss of BMAL1 in skeletal muscle. Thus, these results suggest an important role for skeletal muscle in sleep regulation, implying that *Bmal1*-dependent processes in skeletal muscle may be useful therapeutic targets for sleep disorders. In an effort to investigate the therapeutic potential of muscle *Bmal1*, we examined sleep architecture in wild type mice harboring either the brain or muscle-specific (*Brager et al., 2017*) *Bmal1* transgene. In these mice, transgene expression is in addition to endogenous *Bmal1* expression. Neither baseline sleep nor SWA was significantly altered in brain overexpressed mice (*Figure 4B*). Baseline sleep amount was not significantly altered in muscle- overexpressed mice, however, baseline SWA was significantly reduced (*Figure 4A*), suggesting that overexpressing BMAL1 in the muscle renders mice resistant to sleep loss (*Dijk et al., 1987*). Furthermore, cholinergic neurons of the basal forebrain, which are important for recovery from sleep loss (*Kalinchuk et al., 2015*), are more active in *Bmal1* muscle-overexpressed mice (*Figure 4—figure supplement 1*).

To further investigate recovery from sleep loss, muscle-overexpressed mice were subjected to 24 hr of forced wakefulness by placing mice in a slowly rotating wheel (the 6 hr of forced wakefulness used previously is relatively mild). It is common for mice to exhibit some sleep during prolonged forced-wakefulness paradigms, however, *Bmal1* muscle-overexpressed mice were awake more than WT littermates during these 24 hr of forced wakefulness (*Figure 4C*). Similar to baseline, *Bmal1* muscle-overexpressed mice also had less NREM recovery after forced wakefulness (recovery = sleep gained in recovery/sleep lost during forced wakefulness; *Figure 4C*) and reduced SWA throughout the 72 hr protocol compared to WT mice (*Figure 4D*), despite sleeping less. In addition, waking SWA (a measure of accumulating sleep pressure during extensive durations of sleep loss; *Cajochen et al., 2002*) rose more rapidly and was consistently higher than in WT mice during 24 hr of forced wakefulness (*Figure 4D*). Combined, these results demonstrate that overexpressing *Bmal1* in the skeletal muscle renders mice less sensitive to the effects of sleep loss, portending muscle as a potential therapeutic target for sleep loss.

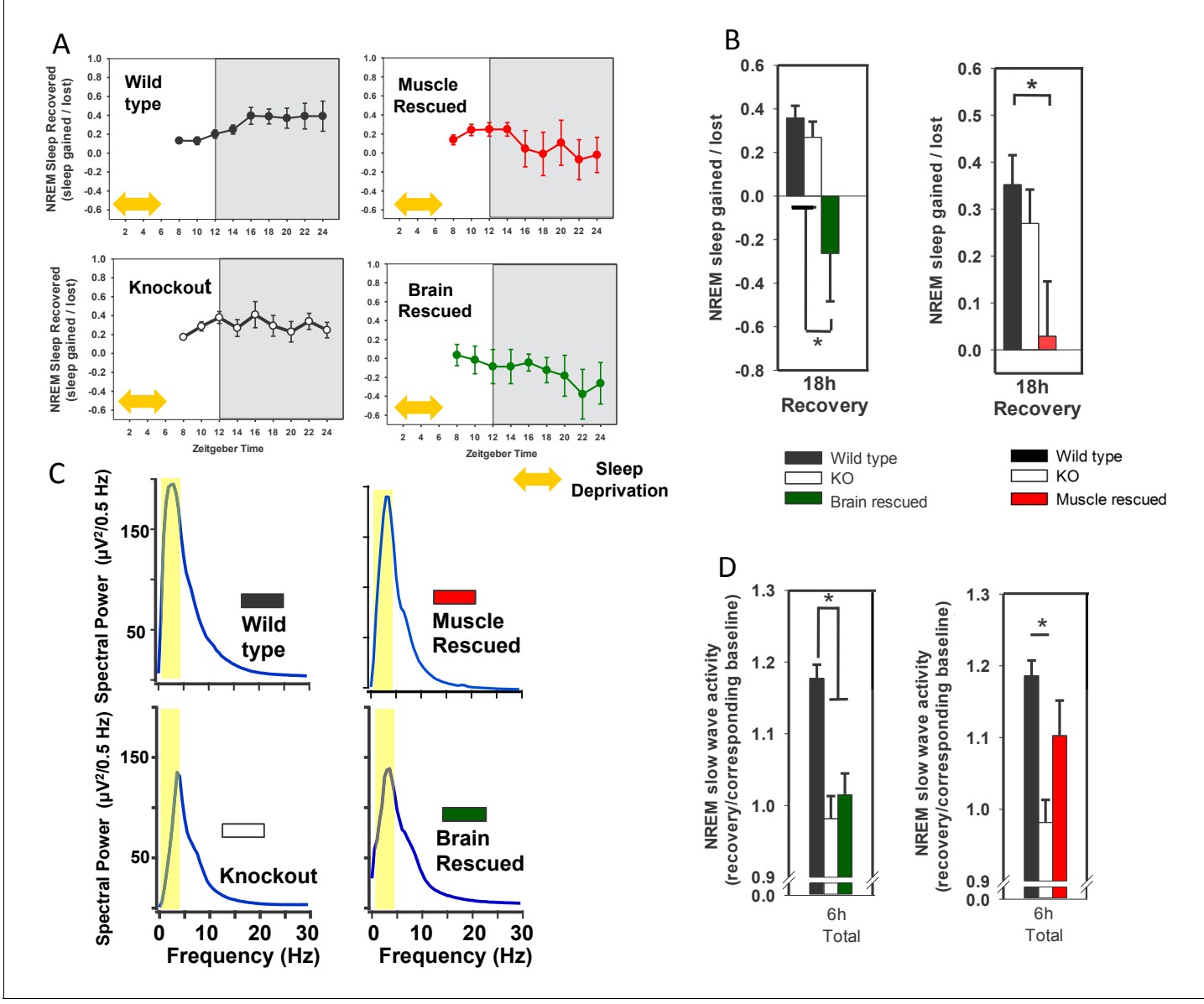

**Figure 2.** Rescuing *Bmal1* in skeletal muscle or brain reduces the amount of NREM sleep recovered after forced wakefulness. Continuous sleep recordings during 18 hr of recovery sleep were obtained from the mice in *Figure 1* after 6 hr of forced wakefulness (A; yellow double arrow = forced wakefulness). The total NREM recovery sleep during this 18 hr period in brain-rescued and muscle-rescued mice was reduced when compared to WT mice (B, ANOVA brain rescued, $F_{(2,28)}$=6.47, p=0.005, p=0.004 Tukey's; ANOVA muscle rescued, $F_{(2,30)}$=5.45, p=0.01 Tukey's). Values in A and B represent sleep time gained after forced wakefulness (calculated using the corresponding interval during undisturbed sleep) as a percentage of total sleep lost (mean ± s.e.m.). The distribution of EEG power during NREM sleep for representative animals from each genotype is shown in (panel C). Slow wave activity (highlighted area) represents power in the 0.5 to 4 Hz frequency band. NREM slow wave activity was reduced in knockout mice following forced wakefulness—compared to WT mice (D, % change over corresponding baseline; p<0.01, Tukey's test). This reduction in slow wave activity was absent when *Bmal1* was rescued in the skeletal muscle (D). Knockout animals in B and D are replicated between graphs to aid in comparison. *p<0.05.

The following figure supplement is available for figure 2:

**Figure supplement 1.** Rescuing *Bmal1* in skeletal muscle or brain reduces the amount of NREM sleep recovered after forced wakefulness.

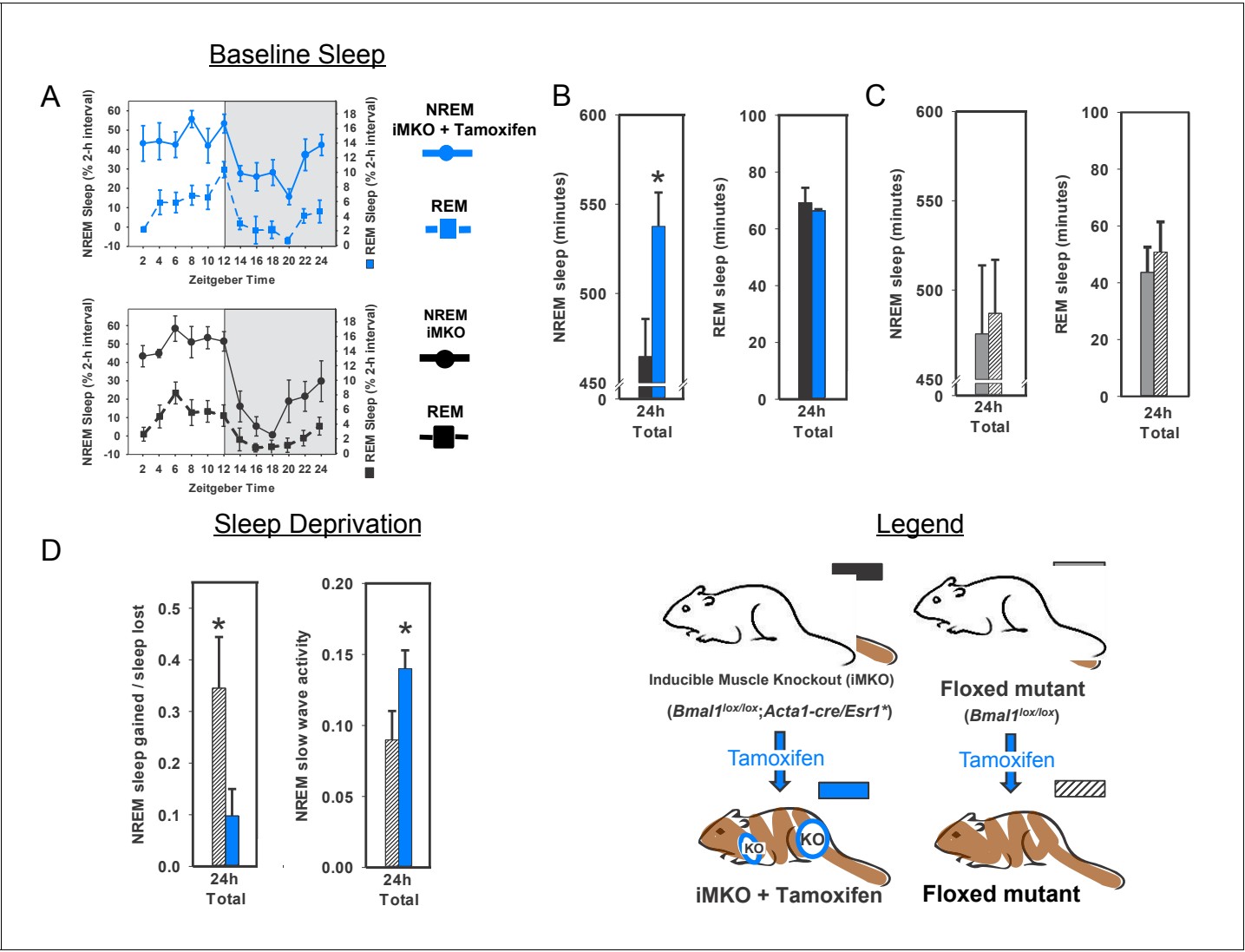

**Figure 3.** Knockout of *Bmal1* in skeletal muscle increases NREM sleep amount and confers resistance to sleep loss. Selective knockout of *Bmal1* in skeletal muscle significantly increased NREM sleep, but not REM sleep, when compared to the same animals prior to tamoxifen treatment (A, B; $t_{(10)}$=2.52, p=0.036). Treatment of floxed mutant control animals with tamoxifen did not significantly alter NREM sleep or REM sleep (C). Following tamoxifen treatment, mice underwent 6 hr of sleep deprivation. Muscle knockout mice also had a significantly altered recovery response to this treatment. Significantly less NREM recovery sleep ($t_{(11)}$=2.44, p=0.033) and increased NREM slow wave activity ($t_{(11)}$=2.2, p=0.05) was observed in muscle knockout mice when compared to tamoxifen-treated floxed mutants (D). n = 6 per group; *p<0.05.

The following figure supplement is available for figure 3:

**Figure supplement 1.** Slow wave activity in inducible muscle knockout mice and controls during undisturbed baseline sleep.

How might peripheral tissues such as muscle influence sleep? The rapidly emerging area of muscle-derived factors on systemic health provides a potential model for our findings. In particular, there are several examples of muscle-derived factors that alter brain processes. Notably, overexpression of PGC-1α (peroxisome proliferator-activated receptor gamma coactivator 1α, gene symbol: *Ppargc1a*) selectively in mouse skeletal muscle reduces the depressive phenotypes induced by stress by preventing plasma kynurenine from reaching the brain (*Agudelo et al., 2014*). Furthermore, PGC-1α activation stimulates release of the muscle-derived peptide irisin into circulation (*Boström et al., 2012*). Plasma irisin, in turn, induces BDNF expression in the hippocampus (*Wrann et al., 2013*). Indeed, *PGC-1α* expression is rhythmic in skeletal muscle (*Liu et al., 2007*)

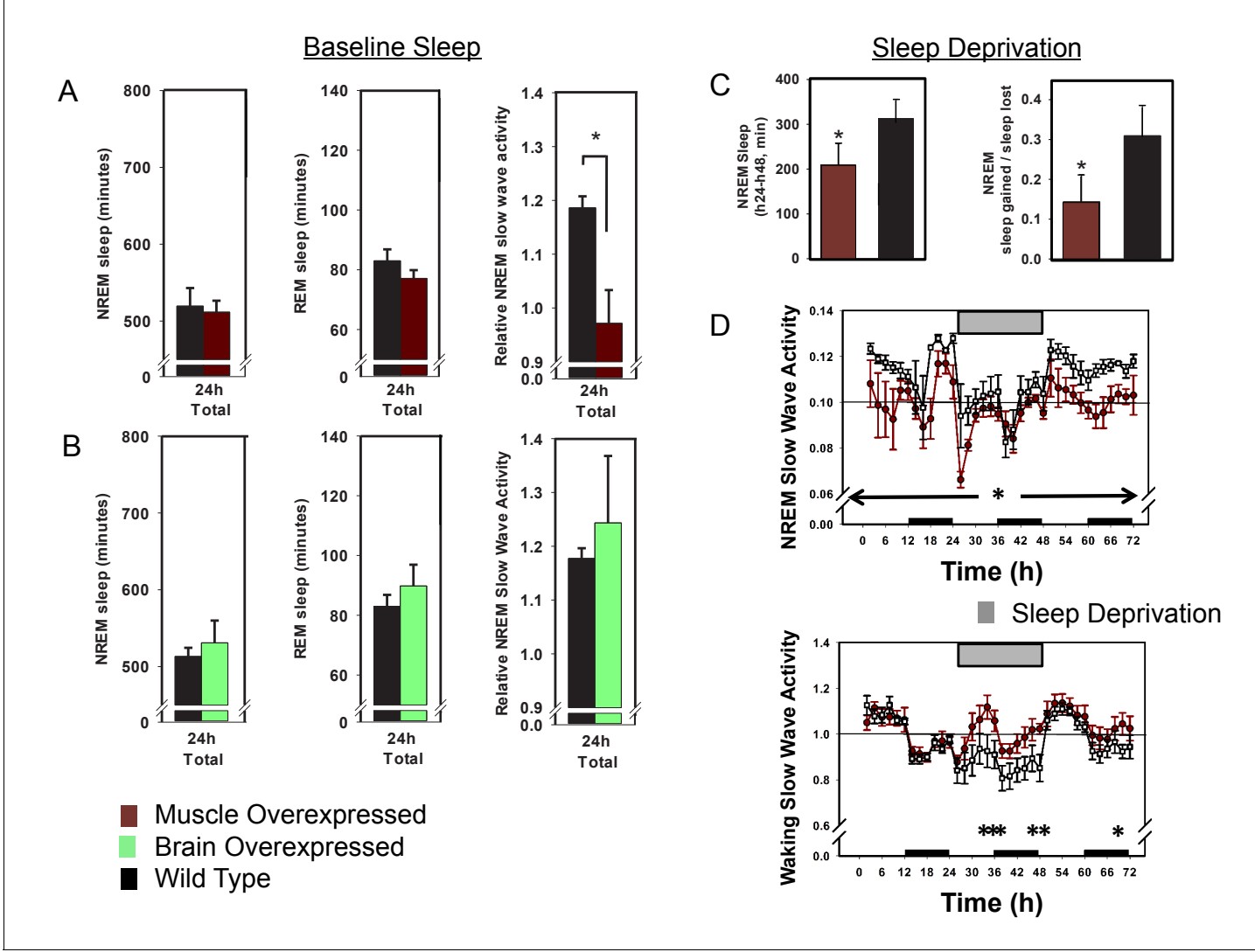

**Figure 4.** Overexpression of *Bmal1* in skeletal muscle confers resistance to sleep loss. 24 hr electroencephalographic recordings were conducted in undisturbed mice overexpressing *Bmal1* in skeletal muscle (**A**) or brain (**B**). No differences in baseline sleep amount were found, however, NREM slow wave activity was significantly reduced in mice overexpressing *Bmal1* in skeletal muscle (**A**). Overexpression of *Bmal1* in skeletal muscle also significantly decreased the NREM-recovery response to one (24 hr) day of forced wakefulness by means of a slowly rotating wheel (**C, D**). It is not unusual for mice to obtain brief amounts of sleep (i.e. micro-sleep) during such an extended regimen of forced wakefulness. This sleep obtained during forced wakefulness was also significantly lower in mice overexpressing *Bmal1* (A, ANOVA main effect of genotype $F_{(1, 96)}$=10.12, p=0.002, main effect of time $F_{(11, 96)}$=4.1, p<0.001; B, NREM sleep amount $t_{(8)}$=1.85, p=0.047, NREM recovery sleep $t_{(8)}$=1.9, p=0.039). NREM slow wave activity (a standard marker of sleep intensity) was consistently lower in *Bmal1* muscle-overexpressed mice throughout the 72 hr protocol (D, repeated-measures ANOVA $F_{(1,8)}$=6.57, p=0.04). In contrast, during forced wakefulness waking slow-wave activity was significantly increased in *Bmal1* muscle-overexpressed mice (D, repeated-measures ANOVA main effect of genotype, $F_{(1,8)}$=5.7, p=0.045). Grey bars indicate forced wakefulness, black bars indicate darkness (active period). Data presented as mean ± s.e.m, n = 16 muscle overexpression, n = 16 littermate controls (muscle), brain overexpression n = 8, littermate controls (brain) n = 11. *p<0.05.

The following figure supplements are available for figure 4:

**Figure supplement 1.** Fos-immunoreactivity (IR) in densely cholinergic areas is increased in *Bmal1* muscle-overexpressed mice.

**Figure supplement 2.** Activity rhythms in *Bmal1* muscle-overexpressed mice are modestly increased when compared to wildtype mice.

**Figure supplement 3.** *Bmal1* muscle-overexpressed mice do not have not altered levels of CLOCK:BMAL1-target genes in the brain or muscle.

raising the possibility that alterations in *Bmal1* expresion/function may alter rhythmic *PGC1α* expression through a change in clock function. However, the studies presented here are not sufficient to determine if the sleep effects of loss/gain of *Bmal1* function in skeletal muscle are via core clock or non-clock mechanisms. Furthermore, whole body deletions of other circadian factors such as *Per1* and *Per2* (*Shiromani et al., 2004*), and *Cry1* and *Cry2* (*Wisor et al., 2008*), do not have similar effects on sleep. Other potential contributors could be related to changes in muscle metabolism as *Bmal1* metabolic phenotypes have been reported in both muscle mouse-lines used here (*Harfmann et al., 2016*; *Brager et al., 2017*).

Recent studies have highlighted that sleep disruptions in humans are associated with peripheral circadian desynchrony (*Cedernaes et al., 2015*; *Schroder and Esser, 2013*). The current study demonstrates that manipulating levels of the circadian transcription factor *Bmal1* specifically in skeletal muscle alters sleep. Moreover, a majority of these effects of *Bmal1* on sleep are not dependent on circadian timing in the brain—dependence on circadian timing in the skeletal muscle remains a possibility. Although it has been established that sleep is important for skeletal muscle function (for a review, see *Chase, 2013*), these investigations are the first to implicate molecular processes within skeletal muscle in signaling sleep regulatory mechanisms in the brain. Studies in our lab are currently underway to determine the nature of the pathway skeletal muscle uses to signal sleep regulatory mechanisms in the brain.

## Materials and methods

### Animals

All mice in the brain lines and muscle rescued/overexpressed lines were maintained on a 12-hr light:12-hr dark schedule throughout the study. Food and water were available *ad libitum*, and animals (10–12 w of age) were individually housed for at least 2 weeks prior to experimental use. All protocols and procedures were approved by the Morehouse School of Medicine Institutional Animal Care and Use Committee.

*Bmal1* brain-rescued and brain overexpressed mice used the tetracycline transactivator (tTA) system, which requires two transgenes for expression of the target gene *Bmal1* (previously reported in *McDearmon et al., 2006*; RRID:MGI:3714773). The promoter sequence of the secretogranin II gene (*Scg2*), which is expressed exclusively in the brain, drives expression of the tetracycline transactivator (*tTA*). The tTA protein binds to the tetracycline operator (*tetO*) sequence and drives expression of *Bmal1* cDNA. The double transgenic mice were crossed onto a *Bmal1* knockout background (RRID: IMSR_JAX:009100) to create brain-rescued mice and were crossed onto a *Bmal1* WT background to create brain overexpressed mice. Breeding was conducted in-house and genotypes were recorded as they became available from breeding. For rescue lines, animals from approximately 20–25 litters comprised the entire dataset. WT littermates obtained from these litters were kept separate for comparisons in each line. KO mice were offspring from independent crosses of heterozygous *Bmal1* KO's.

In situ hybridization studies demonstrate that *Scg2* mRNA is found throughout the brain with the higher expression in the hypothalamus and peak expression in the SCN. Moderate expression of *Scg2* is detected in midbrain and hypothalamic nuclei of the ascending arousal system and in the sleep promoting ventrolateral preoptic area (*Lein et al., 2007*). The mice used in the present study, constructed with a 9.8 kb promoter region of *Scg2*, were characterized previously (*Hong et al., 2007*). Briefly, *Scg2::tTA* mice express the *tTA* transcript broadly in the brain and is enriched in the SCN, when assessed by both an oligo to the *tTA* transcript and by crosses with *tetO*-promotor-linked reporter lines (*Hong et al., 2007*). Furthermore, in situ hybridization in *Scg2::tTA X tetO:: Bmal1-HA* and *tetO::Bmal1*-HA mice demonstrate that *Bmal1* expression is under strict control of the tetracycline transactivator: Hemagglutinin (HA)-tag expression is only detectable in the double transgenic mouse. Both *Bmal1* mRNA and protein are constitutively expressed in the brain of brain-rescued mice. (*McDearmon et al., 2006*). Brain specificity has been demonstrated by western blot which shows an absence of HA-staining in both the muscle and liver of double transgenic mice (*McDearmon et al., 2006*).

*Bmal1* muscle-rescued and muscle-overexpressed mice were generated with the use of a DNA construct consisting of human α-actin-1 promoter sequence positioned upstream of *Bmal1* (RRID:

MGI:3714769; *McDearmon et al., 2006*). The transgenic mice were crossed onto a *Bmal1* knockout background to create muscle-rescued mice, and also crossed onto a *Bmal1* WT background to create brain-overexpressed mice.

The skeletal muscle-specific Cre-recombinase mouse (*Acta1-cre/Esr1*[*], RRID:IMSR_JAX:025750) was generated in house (*McCarthy et al., 2012a*). Breeding with the floxed *Bmal1* mouse (*Bmal1*[lox/lox], The Jackson Laboratory, RRID:IMSR_JAX:007668; *Storch et al., 2007*) generated the inducible muscle knockout mouse. These offspring (*Bmal1*[lox/lox]; *Acta1-cre/Esr1*[*]) allow selective deletion of the bHLH domain of *Bmal1* in skeletal muscle upon tamoxifen administration.

The *Acta1* promoter used for both mouse lines (muscle rescued/overexpressed and muscle KO) is a 2.2 kb sequence directly upstream from the human skeletal actin (*Acta1* gene) translational start site. Proper developmental and tissue-specific expression has been verified previously using an *Acta1::CAT* mouse line. These findings included a demonstrated lack of expression in the brain (*Brennan and Hardeman, 1993*). Specific Cre-dependent excision of a loxP-flanked gene in mouse striated muscle fiber was also demonstrated in mice expressing Cre recombinase under the control of this same promoter (*Miniou et al., 1999*). Transgene expression in the inducible muscle knockout mice (*Acta1-cre/Esr1*\*) used here is constitutive and was not detectable in brain by western blot (*McCarthy et al., 2012b*). A lack transgene expression in the brain has also been demonstrated for the *Acta1::Bmal1-HA* line (*McDearmon et al., 2006*). We verified this finding by western blotting an entire brain hemisphere or gastrocnemeous muscle in mice bred at our facility (*Figure 1—figure supplement 1*). HA-tag was detected in skeletal muscle, but not brain.

## Surgery

EEG and EMG electrodes were implanted in anesthetized mice. A prefabricated head mount (Pinnacle Technologies, KS) was used to position three stainless-steel epidural screw electrodes. The first electrode (frontal—located over the frontal cortex) was placed 1.5 mm anterior to bregma and 1.5 mm lateral to the central suture, whereas the second two electrodes (interparietal—located over the visual cortex and common reference) were placed 2.5 mm posterior to bregma and 1.5 mm on either side of the central suture. The resulting two leads (frontal-interparietal and interparietal-interparietal) were referenced contralaterally. A fourth screw served as a ground. Electrical continuity between the screw electrode and head mount was aided by silver epoxy. EMG activity was monitored using stainless-steel Teflon-coated wires that were inserted bilaterally into the nuchal muscle. The head mount (integrated 2 × 3 pin grid array) was secured to the skull with dental acrylic. Mice were allowed to recover for at least 14 days before sleep recording.

## EEG/EMG Recordings

One week after surgery, mice were moved to the sleep-recording chamber and connected to a lightweight tether attached to a low-resistance commutator mounted over the cage (Pinnacle Technologies). This enabled complete freedom of movement throughout the cage. Except for the recording tether, conditions in the recording chamber were identical to those in the home cage. Breeding was conducted in-house, and genotypes were recorded as they became available from breeding. Recordings from approximately 20–25 litters makup the dataset. All wildtype and KO littermates resulting from a litter, up to a maximum of 50%, were recorded with tissue-specific knockout/rescue mice. Mice were allowed a minimum of 7 additional days to acclimate to the tether and recording chamber. Recording of EEG and EMG waveforms began at zeitgeber time (ZT) 0 (light onset). Data acquisition was performed on a personal computer running Sirenia Acquisition software (Pinnacle Technologies), a software system designed specifically for polysomnographic recording in rodents. EEG signals were low-pass filtered with a 40 Hz cutoff and collected continuously at a sampling rate of 400 Hz. After collection, all waveforms were classified by a trained observer (using both EEG leads and EMG) as wake (low-voltage, high-frequency EEG; high-amplitude EMG), NREM sleep (high-voltage, mixed-frequency EEG; low- amplitude EMG) or rapid eye movement (REM) sleep (low-voltage EEG with a predominance of theta activity [6–10 Hz]; very low amplitude EMG). EEG epochs determined to have artifact (interference caused by scratching, movement, eating, or drinking) were excluded from analysis. Recordings where artifact comprised more than 5% of total recording time were excluded from analysis. Analysis of NREM delta power and NREM spectral distribution was accomplished by applying a fast Fourier transformation to raw EEG waveforms. Only epochs

classified as NREM sleep were included in this analysis. Delta power was measured as spectral power in the 0.5 to 4 Hz frequency range and expressed as a percentage of total spectral power in the EEG signal (0.5–100 Hz) during that time period.

## Forced wakefulness

Homeostatic challenge: Six-hour forced wakefulness was conducted in all mouse lines (*Figure 2*). Following a 24 hr baseline recording, mice were sleep deprived during the first 6 hr of the light phase (ZT 0–6) by gentle handling (introduction of novel objects into the cage, tapping on the cage and when necessary delicate touching) and allowed an 18-hr recovery opportunity (ZT 6–0).

Twenty-four-hour forced wakefulness: Following a 24 hr baseline recording, *Bmal1* muscle-overexpressed lines and WT littermates were moved to a slowly rotating wheel (nine inches in diameter; 1 rpm) adjacent to the recording cage (*Figure 3*). Mice were confined to this wheel for 24 hr beginning at ZT 0 (lights on) during which time they had free access to food and water. Following sleep deprivation, animals were returned to the baseline recording cage and EEG acquisition was continued for a 24 hr recovery opportunity.

## Fos- and choline acetyltransferase (ChAT) immunoreactivity (IR)

Mice were sacrificed by $CO_2$ inhalation at ZT 6 (ZT0 represented lights-on under LD) after 6 hr of forced wakefulness. Brains were immersion-fixed in 4% paraformaldehyde for 24 hr then sunk in 30% sucrose (24 hr at 4°C). Cryostat sections (40µm-thick) were incubated with a rabbit polyclonal IgG antibody (c-fos; Santa Cruz Biotechnology, Santa Cruz, CA; chicken polyclonal IgY antibody [ChAT]; Novus Biologicals, Littleton, CO) and immunoreactivity was visualized using Vectastain Elite ABC kit with 3,3-diaminobenzidine tetrahydrochloride (DAB) as chromogen (Vector Labs, Burlingame, CA). Sections were mounted with permount, and Fos expression was quantified using ImageJ (National Institutes of Health, Bethesda, MD). Counts of immunostained nuclei were undertaken in the mid-to-posterior region of each brain site.

## Quantitative PCR

Methods were described previously (*McCarthy et al., 2012b*). Briefly, brain (hypothalamic) and skeletal muscle (gastrocnemius) tissues were collected from mice sacrificed at ZT5 and ZT17 (ZT12 was lights-off under a 12 hr: 12 hr light:dark cycle); these are the mid-points of peak and trough *Bmal1* gene expression in skeletal muscle (*McCarthy et al., 2007*). Total RNA was extracted from frozen brain and skeletal muscle using Trizol (Invitrogen, Carlsbad, CA) and diluted to 0.1 mg/ml. Samples were converted from RNA to cDNA with Applied Biosystems RT-PCR kit reagents according to the manufacturer's instruction. Real-time PCR assays were performed using the comparative amplification detection threshold of target gene expression (CT) method. mRNA levels detected with SYBR Green (Bio-Rad; Hercules, CA) were measured by determining the cycle number at which CT was reached. In each sample, CT was normalized to *Gapdh* expression (ΔCT) performed on the same plate. Normalized gene expression (ΔΔCT) for each gene with respect to genotype and time point was computed with Bio-Rad CFX Manager.

## Western blotting

Brain hemispheres were homogenized in a microfuge tube using a pellet pestle in 700 ul lysis buffer (20 mM Hepes pH 7.6, 400 mM NaCl, 1 mM EDTA, 5 mM NaF, 0.3% Triton-X 100, 5% glycerol, 1 mM DTT, 250 nM PMSF, and complete protease inhibitor mix (Sigma), and tumbled at 4C overnight prior to quantification and loading on the gel. Muscle proteins were extracted similarly, except 300 µl to lysis buffer was used, and the lysis buffer contained 1% Triton X-100% and 10% glycerol. Protein concentrations were determined using a BCA assay kit (Pierce), and separated on an Any-kDa minigel (BioRad). Western blot was performed with anti-HA-HRP conjugated monoclonal antibody (Roche) and anti-Gapdh (Santa Cruz).

## Behavioral phenotyping

### Wheel running

Male mice were provided with continuous access to running wheels interfaced to a ClockLab data acquisition system (Coulbourn Instuments, Whitehall, PA) for two weeks under LD followed by 2

weeks under constant darkness (DD; n = 8/genotype). Data were collected in 10 min bins. Nighttime activity onset (designated as zeitgeber time [ZT] 12 under LD and circadian time [CT] 12 under DD) was defined by the initial 10 min period that: (1) exceeded 10% of the maximum activity for the day, (2) was preceded by at least 4 hr of inactivity, and (3) was followed by at least 60 min of sustained activity. Nighttime activity offset was defined as the final period of nighttime activity that: (1) was preceded by at least 60 min of sustained activity and (2) was followed by at least 4 hr of inactivity. Alpha (measured in h) represented the time between nighttime activity onset and offset. Rhythm period was determined from extrapolation of the least squares line through activity onsets on days 3–14 under LD and DD. The duration (bout) and intensity (count) of wheel running under LD and DD were also measured.

## Beam breaks

A SmartHomeCage data acquisition system (AfaSci, Redwood City, CA) equipped with a multi-layer 3D array of infrared sensors was placed around the home cages of male mice (n = 7–8/genotype) to measure gross and fine motor movements under LD (following entrainment) and DD (third full 24 hr period, CT time derived from each individual animal's rhythm period). Behavioral parameters included the number of spontaneous movements (counts), % activity in 1 hr blocks, distance travelled in home cage, and minutes of gross (ambulatory) and fine (grooming) motor movements.

## Statistics

Sleep data were analyzed using one-way analysis of variance (ANOVA), repeated-measures ANOVA or Student's *t*. Significance was defined as p≤0.05. *Post hoc* analysis was conducted using Tukey's HSD method or student *t*-test where indicated. Tukey's HSD method uses the studentized range statistic and maintains family-wise error-rate at 0.05. An appropriate sample size of 5 was predicted with Type I error rate of 0.05 and Type II error rate of 0.2. Standard deviation and mean difference were estimated as 46.4 and 100 min, respectively, based on the existing literature (*Laposky et al., 2005*) and previous studies in our lab. Sample sizes (biological replicates) for each experiment are indicated in the figure legends.

## Supplementary materials

### Overexpression of *Bmal1* in the skeletal muscle increases Fos-IR in cholinergic neurons

The finding that *Bmal1* muscle overexpression altered brain EEG led us to investigate whether the manipulation had other influences on brain phenotypes that are related to sleep-wake processes. Therefore, we determined the effects of *Bmal1* muscle overexpression on neuronal activation in sleep regulatory regions. FOS-immunoreactivity (IR) was measured in the following brain areas: (1) forebrain (nucleus accumbens [NAc/s], basal forebrain [BF], ventral pallidum [VP]); (2) hypothalamus (ventral lateral [VLPO] and medial preoptic [MPOA] areas, suprachiasmatic nucleus [SCN]); (3) thalamus (lateral habenula [LAh]); and (4) hindbrain (ventral [VTA], pedunculopontine [PPT], and laterodorsal [LDT] tegmental areas (S1). Brains were collected at midday under basal (undisturbed) conditions or immediately after a 6 hr of sleep deprivation.

FOS-IR was increased in the forebrain, midbrain, and hindbrain of *Bmal1* muscle-overexpressed mice relative to WTs (BF: $F_{1,16}$=26.6; p<0.001, univariate ANOVA; Hb: $F_{1,16}$=75.6; p<0.001; PPT: $F_{1,16}$=31.0; p<0.001; LDT: $F_{1,16}$=34.5; p<0.001). There were also significant increases in FOS-IR in the forebrain, midbrain and hindbrain (NAc: $F_{1,16}$=15.5; p=0.001; BF: $F_{1,16}$=6.8; p=0.01; PPT: $F_{1,16}$=12.6; p=0.004; Hb: $F_{1,16}$=26.6; p<0.001; MpOA; $F_{1,16}$=15.6; p=0.001) of sleep-deprived mice. Most notably, there were no main effects for genotype or treatment in the SCN. This observation indicates that disrupted circadian timing is not responsible for sleep phenotypes in *Bmal1* muscle-overexpressed mice.

Coincidentally, the brain areas that had increases in FOS-IR in response to *Bmal1* muscle overexpression—the basal forebrain, lateral habenula, and pedunculopontine tegmentum—are known to be arousal promoting and have extensive cholinergic tone. We double-labeled with ChAT (enzyme necessary for acetylcholine synthesis) to determine if cholinergic neurons are FOS-positive. Our findings indicate that increases in Fos-IR of *Bmal1* muscle-overexpressed mice occurred in cholinergic neurons (BF: $F_{1,20}$=5.6; p=0.02; Hb: $F_{1,20}$=14.8; p=0.002; PPT: $F_{1,20}$=13.5; p=0.003; S1).

## Overexpression of *Bmal1* in the skeletal muscle does not cause circadian impairments in wheel running

The patterns of circadian wheel-running activity have been reported in the majority of mouse models reported here (*McDearmon et al., 2006*; *Wisor et al., 2008*); however, this wheel-running activity is not available for *Bmal1* muscle-overexpressed mice. Therefore, we investigated behavioral circadian rhythms in *Bmal1* muscle-overexpressed mice and WT littermates.

*Bmal1* muscle-overexpressed mice entrained to LD and had a nighttime activity onset of 12.2 ± 8.4 min after lights-off. WTs entrained to LD with a nighttime activity onset of 5.4 ± 3.4 min after lights-off (*onset*: $F_{1,14}=0.77$; p=0.39, one-way ANOVA). The length of the nighttime activity period averaged 11.0 ± 1.1 hr for *Bmal1* muscle- overexpressed mice and 10.7 ± 0.6 hr for WTs ($F_{1,14}=0.06$; p=0.81, one-way ANOVA). The average duration of a wheel running bout was 176.2 ± 52.3 min for *Bmal1* muscle-overexpressed mice and 131.6 ± 19.4 min of wheel running for WTs (*duration*: $F_{1,14}=1.3$; p=0.27; *intensity*: $F_{1,14}=0.49$; p=0.41, one-way ANOVA). Overexpression of *Bmal1* in the skeletal muscle did not change the endogenous rhythm of wheel running. Rhythm period for *Bmal1* muscle-overexpressed mice averaged 23.7 ± 0.1 hr compared with 23.8 ± 0.2 hr for WTs ($F_{1,13}=0.90$; p=0.36, one-way ANOVA). The duration of the subjective night was also unaffected in *Bmal1* muscle overexpressed-mice compared with WTs (10.4 ± 1.1 hr vs. 9.1 ± 0.9 hr, respectively; $F_{1,13}=0.18$; p=0.68, one-way ANOVA).

## Acknowledgements

The authors would like to express our sincere gratitude to Peter MacLeish and Gianluca Tosini for their critical analyses of this work. We would like to thank Zach W Hall and India Nichols for providing critical review of this manuscript, and Karrie Fitzpatrick and Constantin Chikando for technical support. Funding support provided by: R01 NS078410 (KNP), U54 NS060659 (KNP), F32 HL116077 (AJB), P50 HL117929 (KNP), G12 MD007602 (JCE, JPD), U54 NS083932 (JCE, JPD, KNP), SC1 GM109861 (JPD), T32 HL007609 (CLG), P50 MH074924 (JST). JST is an Investigator in the Howard Hughes Medical Institute.

## Additional information

### Competing interests

JST: Reviewing editor, *eLife*. The other authors declare that no competing interests exist.

### Funding

| Funder | Grant reference number | Author |
| --- | --- | --- |
| National Institute on Minority Health and Health Disparities | G12 MD007602 | J Christopher Ehlen<br>Jason P DeBruyne |
| National Institute of Neurological Disorders and Stroke | U54 NS083932 | J Christopher Ehlen<br>Jason P DeBruyne<br>Ketema N Paul |
| National Heart, Lung, and Blood Institute | F32 HL116077 | Allison J Brager |
| National Academy of Sciences | National ResearchCouncil Research Associateship | Allison J Brager |
| National Heart, Lung, and Blood Institute | T32 HL007609 | Cloe L Gray |
| National Institute of General Medical Sciences | SC1 GM109861 | Jason P DeBruyne |
| National Institute of Mental Health | P50 MH074924 | Joseph S Takahashi |
| Howard Hughes Medical Institute | | Joseph S Takahashi |
| National Institute of Neurological Disorders and Stroke | R01 NS078410 | Ketema N Paul |

| National Institute of Neurological Disorders and Stroke | U54 NS060659 | Ketema N Paul |

The funders had no role in study design, data collection and interpretation, or the decision to submit the work for publication. The opinions or assertions contained herein are the private views of the author, and are not to be construed as official, or as reflecting true views of the Department of the Army or the Department of Defense.

## Author contributions

JCE, AJB, Conceptualization, Formal analysis, Investigation, Methodology, Writing—original draft, Writing—review and editing; JB, Investigation, Methodology; LP, Formal analysis, Investigation; CLG, Investigation, Methodology, Writing—original draft, Writing—review and editing; JPD, Resources, Formal analysis, Investigation, Methodology, Writing—original draft, Writing—review and editing; KAE, Resources, Investigation, Methodology, Writing—original draft, Writing—review and editing; JST, Resources, Writing—original draft, Writing—review and editing; KNP, Conceptualization, Resources, Funding acquisition, Methodology, Writing—original draft, Writing—review and editing

## Author ORCIDs

J Christopher Ehlen, http://orcid.org/0000-0003-3223-9262

Karyn A Esser, http://orcid.org/0000-0002-5791-1441

Joseph S Takahashi, http://orcid.org/0000-0003-0384-8878

Ketema N Paul, http://orcid.org/0000-0003-0226-9559

## Ethics

Animal experimentation: All procedures involving animals were approved by the Morehouse School of Medicine institutional animal care and use committee, protocol reference number 15-17. Animal studies conformed to recommendations published in the Guide for the Care and Use of Laboratory Animals of the National Institutes of Health.

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
