## [Decision Letter]

Thank you for submitting your article "*Bmal1* function in skeletal muscle regulates sleep" for consideration by *eLife*. Your article has been reviewed by three peer reviewers, one of whom, J Ptáček (Reviewer #1), is a member of our Board of Reviewing Editors, and the evaluation has been overseen by a Senior Editor. The following individuals involved in review of your submission have agreed to reveal their identity: John B Hogenesch (Reviewer #2) and Katja A Lamia (Reviewer #3).

The reviewers have discussed the reviews with one another and the Reviewing Editor has drafted this decision to help you prepare a revised submission.

Summary:

In Ehlers et al., the authors study the role of *Bmal1* in regulating sleep and recovery from sleep debt. Earlier, it was noted that *Bmal1* null mice, like most circadian clock null mice, have sleep abnormalities including increased total sleep and NREM sleep intensity and reduced recovery to sleep loss. The authors rescued *Bmal1* function in the brain with a *Scg2::tTA* system, but while this rescued circadian function, surprisingly, it had no effect on sleep. Next, they restored *Bmal1* function to skeletal muscle using a *Acta1* driven *Bmal1* construct in a constitutive fashion and showed that *Bmal1* in muscle is sufficient to restore normal NREM sleep amount. They went on to show similar results for recovery from sleep deprivation. To demonstrate the necessity of *Bmal1* function in skeletal muscle, they generated muscle-specific *Bmal1* knockout mice and showed sleep phenotypes consistent with muscle *Bmal1* function being important for sleep. They went on to overexpress *Bmal1* in skeletal muscle, sleep deprived animals, and showed further resistance to sleep deprivation, demonstrating that *Bmal1* is gene dose limiting in mediating resistance sleep deprivation.

The findings are supported by the data and will be of interest to a broad scientific audience. The paper could be strengthened by more thorough description and discussion of their data in relation to other studies. Some of the conclusions should be clarified with respect to possible underlying direct or indirect mechanisms and in relation to other findings. Finally, some of the data presentation should be clarified.

Despite these technical concerns, I feel this is interesting and worth reporting but that it will be necessary for the authors to acknowledge potential problems noted above to be sufficiently conservative in interpretation.

Essential revisions:

This is a very interesting idea and the data is strongly suggestive. It is clear that exercise improves subsequent sleep quality and it is interesting to speculate that this might result from some humoral factor released from muscle. The main concern is that the authors are not sufficiently careful in the interpretation of their findings. They used *Scg2-Bmal1* to rescue the brain expression. Obviously the expression of *Scg2* (levels, cell types, etc.) are somewhat different from that of endogenous *Bmal1*, although the circadian rhythm phenotype could be rescued. However, little is known about sleep regulation and all of the brain regions that might be involve. So even if *Scg2-Bmal1* could not rescue the *Bmal1* KO sleep phenotype, it is overstated that brain-specific expression of *Bmal1* could not rescue the sleep phenotype. Perhaps the relevant brain region (and specific cell types) do not have sufficiently high expression of the exogenous *Bmal1* to rescue the phenotype.

Similarly, they used *Acta1-Bmal1* as the model of muscle specific rescue. Although *Bmal1* in this model is supposed to be restricted, it is impossible to exclude the expression of this promoter in some specific area or cell types in the brain that might be critically relevant for rescue of the phenotype.

1) The authors speculate that a *Bmal1* target gene involved in regulation of a circulatory factor mediates the sleep effects seen in muscle perturbation of *Bmal1* function. This seems highly likely. Does this imply it's likely to be a circadian-clock regulated gene or a non-circadian target of *Bmal1*? This point may merit discussion.

2) The promoters used here should be described more clearly. E.g. Where is *Scg2* expressed in the brain, broadly or discretely? Constitutively or in a circadian fashion? Similarly, while the *Acta1* promoter used is supposed to be quite clean, was this verified by ICC or immunoblot? Expression of transgenes can 'leak' over time.

3) The kyurenine hypothesis is interesting, but as far as I can tell, the aminotransferase thought to be responsible, Ccbl1, is more likely to be a Ror (driven by Pgc1a) than *Bmal1* target. (Ccbl1 also shares phase with *Bmal1* in liver, opposite from canonical *Bmal1* targets such as Dbp and Nr1d1.)

5) There seems to be a correlation between reduced locomotor activity, altered temporal organization of activity, and reduced NREM sleep and altered temporal organization of NREM sleep patterns. Could altered levels of physical activity contribute to differences in sleep structure?

6) The authors should more carefully describe the phenotypes to clearly state that the diurnal patterns of sleep are not restored by rescuing *Bmal1* expression in skeletal muscle. Perhaps Figure 1 could include quantification of the light and dark phase NREM and REM sleep. This is reminiscent of the earlier report showing that in a *Bmal1*KO background rescue of *Bmal1* in brain restored diurnal activity patterns but not the overall amount of activity while rescue in muscle restored levels of activity but not consolidation (McDearmon et al., 2006). That study also showed that rescuing *Bmal1* in muscle restored body weight of *Bmal1* KO mice. Could there be an overarching metabolic change that contributes to both body weight restoration and sleep propensities?

7) The statement in the Abstract that "[…] most of these phenotypes could be reproduced or rescued by knocking out or restoring BMAL1 exclusively in the skeletal muscle […]" is too strong. This should be qualified with a statement about rescuing the overall levels but not the altered timing of sleep.

8) More detail should be provided with respect to the tissue specificity of the mouse lines used. The *Scg2::tTa* is limited to the SCN and several other regions; this should be clarified to avoid the implication that rescue of *Bmal1* expression throughout the brain has no effect on sleep. Is it known that *Acta1-cre/Esr1** line does not express Cre in brain regions that are important for modulating sleep?

9) In Figure 1 and Figure 2, "wildtype" and "knockout" data are presented with both the muscle-rescued line and the brain-rescued line but it is unclear whether these are the same data from an independent colony or whether the "knockout" mice are littermates of the brain-rescued and muscle-rescued animals.

10) In Figure 2, it is unclear what exactly is shown in panels A and B. It seems to be a calculation based on comparison to the corresponding interval during undisturbed sleep for the same genotype or the same animal but could this artificially alter the value since the undisturbed diurnal sleep patterns are quite different for the different genotypes? It might be better to present the "baseline" and the "recovery" data together to get a clear picture of how the genetic changes affect recovery from sleep deprivation, as in Laposky et al., 2005 demonstrating effects of ubiquitous *Bmal1* loss on sleep.

---

## [Author Response]

*Essential revisions:*

*[…] 1) The authors speculate that a Bmal1 target gene involved in regulation of a circulatory factor mediates the sleep effects seen in muscle perturbation of Bmal1 function. This seems highly likely. Does this imply it's likely to be a circadian-clock regulated gene or a non-circadian target of Bmal1? This point may merit discussion.*

We appreciate the reviewer’s question regarding potential clock vs. non-clock mechanisms. We have attempted to address this in our discussion of potential mechanisms in paragraphs 8 and 9. In our opinion the current data do not provide sufficient evidence and determining the involvement of the clock will depend on identification of the mechanistic link between muscle and brain. Therefore, we have integrated this discussion with our discussion of muscle-derived factors. A complete description of these changes can be found in 3) below.

It is important to point out that whole-body deletions of other canonical circadian clock genes do not have similar effects on sleep. The following sentence was added to paragraph 8 of the manuscript: “Furthermore, whole body deletions of other circadian factors such as Per1 and Per2 (Shiromani et al., 2004), and Cry1 and Cry2 (Wisor et al., 2008), do not have similar effects on sleep.”-

*2) The promoters used here should be described more clearly. E.g. Where is Scg2 expressed in the brain, broadly or discretely? Constitutively or in a circadian fashion? Similarly, while the Acta1 promoter used is supposed to be quite clean, was this verified by ICC or immunoblot? Expression of transgenes can 'leak' over time.*

We have included Figure 1—figure supplement 1 to demonstrate the appropriate tissue-specific expression of the transgene in the *Acta1::Bmal1-HA* line. In addition, the following text has been added to the Material and methods section:*“In situ* hybridization studies demonstrate that *Scg2* mRNA is found throughout the brain with the higher expression in the hypothalamus and peak expression in the SCN. […] Brain specificity has been demonstrated by western blot which shows an absence of HA-staining in both the muscle and liver of double transgenic mice (McDearmon, 2006*).”*

“The *Acta1* promoter used for both mouse lines (muscle rescued/overexpressed and muscle KO) is a 2.2 kb sequence directly upstream from the human skeletal actin (*Acta1* gene) translational start site. […] We verified this finding by western blotting an entire brain hemisphere or gastrocnemeous muscle in mice bred at our facility (Figure 1—figure supplement 1). HA-tag was detected in skeletal muscle, but not brain.”

*3) The kyurenine hypothesis is interesting, but as far as I can tell, the aminotransferase thought to be responsible, Ccbl1, is more likely to be a Ror (driven by Pgc1a) than Bmal1 target. (Ccbl1 also shares phase with Bmal1 in liver, opposite from canonical Bmal1 targets such as Dbp and Nr1d1.)*

Thank you for pointing this out. We have altered the text to clarify our hypothesis and address comment 1). The text in paragraph 8 and 9 now reads: “How might peripheral tissues such as muscle influence sleep? […] Furthermore, whole body deletions of other circadian factors such as *Per1* and *Per2* (Shiromani et al., 2004), and *Cry1* and *Cry2* (Wisor et al., 2008), do not have similar effects on sleep.”

*5) There seems to be a correlation between reduced locomotor activity, altered temporal organization of activity, and reduced NREM sleep and altered temporal organization of NREM sleep patterns. Could altered levels of physical activity contribute to differences in sleep structure?*

A correlation between decreased activity and altered temporal organization is seen in whole-body *Bmal1* KO’s (Pendergast et al., 2008) and WT levels of activity are associated with WT-like sleep structure in the muscle-KO line (Harfmann et al., 2016). However, the patterns observed in the rescue lines are not consistent with this hypothesis. Locomotor activity remains low in brain-rescued mice and sleep structure is partially restored, whereas activity levels are partially restored in the muscle-rescued mice without a restoration of sleep architecture (McDearmon et al., 2006). Based on these data, it still isn’t clear if differences in physical activity contributes to differences in sleep architecture.

*6) The authors should more carefully describe the phenotypes to clearly state that the diurnal patterns of sleep are not restored by rescuing Bmal1 expression in skeletal muscle. Perhaps Figure 1 could include quantification of the light and dark phase NREM and REM sleep. This is reminiscent of the earlier report showing that in a Bmal1KO background rescue of Bmal1 in brain restored diurnal activity patterns but not the overall amount of activity while rescue in muscle restored levels of activity but not consolidation (McDearmon et al., 2006). That study also showed that rescuing Bmal1 in muscle restored body weight of Bmal1 KO mice. Could there be an overarching metabolic change that contributes to both body weight restoration and sleep propensities?*

We have altered paragraph 3 to address this first issue. The text now reads: “To begin this investigation of other tissues we chose mice harboring a transgene that restores *Bmal1* specifically in skeletal muscle, but does not restore circadian behavior (i.e., muscle rescued; *Acta1::Bmal1-HA*).”

“These experiments demonstrate that restoring *Bmal1* in the skeletal muscle of otherwise *Bmal1*-deficient mice is sufficient to restore normal NREM sleep amount, independently of *Bmal1* expression in the brain. The diurnal rhythm in sleep amount, however, is not restored (Figure 1).”

We have begun to address the issue of metabolic changes in our recent publication using muscle-overexpressed mice (Brager et al., 2017). We found no baseline differences in: body composition, respiratory exchange ratio, energy expenditure, total activity counts, and ex vivo insulin sensitivity in skeletal muscle, between muscle-overexpressed and WT mice. We did find that muscle-overexpressed mice had lower energy expenditure during treadmill running and reduced insulin sensitivity following sleep deprivation (Brager et al., 2017). This indicates that there are few metabolic changes in the muscle-overexpressed mice, a line that exhibits significant differences in SWA; however, differences in body composition and glucose tolerance were recently reported in muscle KO mice (Harfmann et al., 2016) and this line displayed increased sleep amount and SWA. Thus, it is possible that metabolic phenotypes contribute to the sleep phenotypes described here.

We have added the following to paragraphs 8 to address this: “Other potential contributors could be related to changes in muscle metabolism as *Bmal1* metabolic phenotypes have been reported in both muscle mouse-lines used here (Harfmann et al., 2016; Brager et al., 2017).”

*7) The statement in the Abstract that "[…] most of these phenotypes could be reproduced or rescued by knocking out or restoring* BMAL1 *exclusively in the skeletal muscle […]" is too strong. This should be qualified with a statement about rescuing the overall levels but not the altered timing of sleep.*

We have rewritten this sentence in the Abstract: “Surprisingly, most sleep-amount, but not sleep-timing, phenotypes could be reproduced or rescued by knocking out or restoring BMAL1 exclusively in skeletal muscle, respectively.”

*8) More detail should be provided with respect to the tissue specificity of the mouse lines used. The Scg2::tTa is limited to the SCN and several other regions; this should be clarified to avoid the implication that rescue of Bmal1 expression throughout the brain has no effect on sleep. Is it known that Acta1-cre/Esr1* line does not express Cre in brain regions that are important for modulating sleep?*

The transgene expression-patterns exhibited in these lines is addressed in item 2).

*9) In Figure 1 and Figure 2, "wildtype" and "knockout" data are presented with both the muscle-rescued line and the brain-rescued line but it is unclear whether these are the same data from an independent colony or whether the "knockout" mice are littermates of the brain-rescued and muscle-rescued animals.*

We have added the following text to the Material and methods to clarify: “For rescue lines, animals from approximately 20-25 litters comprised the entire dataset. WT littermates obtained from these litters were kept separate for comparisons in each line. KO mice were offspring from independent crosses of heterozygous *Bmal1* KO’s”.

*10) In Figure 2, it is unclear what exactly is shown in panels A and B. It seems to be a calculation based on comparison to the corresponding interval during undisturbed sleep for the same genotype or the same animal but could this artificially alter the value since the undisturbed diurnal sleep patterns are quite different for the different genotypes? It might be better to present the "baseline" and the "recovery" data together to get a clear picture of how the genetic changes affect recovery from sleep deprivation, as in Laposky et al., 2005 demonstrating effects of ubiquitous Bmal1 loss on sleep.*

A major problem with interpreting the recovery response to a homeostatic challenge is controlling for the amount of sleep lost. This is especially true when comparing different strains where the amount of sleep lost during sleep deprivation can be different. To control for these differences, normalizing for the amount of lost sleep (by expressing sleep recovered as a percentage of sleep lost) is a standard practice in the sleep field. However, the pattern of sleep recovery does differ between genotypes and we agree that presenting the recovery day, as requested, will allow visualization of these differences. We have presented the data in both formats, the graphs requested are now included in the Figure 2—figure supplement 1.

Brager AJ, Heemstra L, Bhambra R, Ehlen JC, Esser KA, Paul KN, Novak CM (2017) Homeostatic effects of exercise and sleep on metabolic processes in mice with an overexpressed skeletal muscle clock. Biochimie 132:161-165.

Brennan KJ, Hardeman EC (1993) Quantitative analysis of the human α-skeletal actin gene in transgenic mice. J Biol Chem 268:719-725.

Harfmann BD, Schroder EA, Kachman MT, Hodge BA, Zhang X, Esser KA (2016) Muscle-specific loss of *Bmal1* leads to disrupted tissue glucose metabolism and systemic glucose homeostasis. Skelet Muscle 6:12.

Hong HK, Chong JL, Song W, Song EJ, Jyawook AA, Schook AC, Ko CH, Takahashi JS (2007) Inducible and reversible Clock gene expression in brain using the tTA system for the study of circadian behavior. PLoS Genet 3:e33.

Lein, E.S. et al. (2007) Genome-wide atlas of gene expression in the adult mouse brain, Nature 445: 168–176.

McCarthy JJ, Srikuea R, Kirby TJ, Peterson CA, Esser KA (2012) Inducible Cre transgenic mouse strain for skeletal muscle-specific gene targeting. Skelet Muscle 2:8.

McDearmon EL, Patel KN, Ko CH, Walisser JA, Schook AC, Chong JL, Wilsbacher LD, Song EJ, Hong HK, Bradfield CA, Takahashi JS (2006) Dissecting the functions of the mammalian clock protein BMAL1 by tissue-specific rescue in mice. Science 314:1304-1308.

Miniou P, Tiziano D, Frugier T, Roblot N, Le MM, Melki J (1999) Gene targeting restricted to mouse striated muscle lineage. Nucleic Acids Res 27:e27.

Pendergast JS, Nakamura W, Friday RC, Hatanaka F, Takumi T, Yamazaki S (2009) Robust food anticipatory activity in BMAL1-deficient mice. PLoS One 4:e4860.

Shiromani PJ, Xu M, Winston EM, Shiromani SN, Gerashchenko D, Weaver DR (2004) Sleep rhythmicity and homeostasis in mice with targeted disruption of mPeriod genes. Am J Physiol. Regul. Integr. Comp Physiol. 287(24): R47—R57

Wisor JP, Pasumarthi RK, Gerashchenko D, Thompson CL, Pathak S, Sancar A, Franken P, Lein ES, Kilduff TS (2008) Sleep deprivation effects on circadian clock gene expression in the cerebral cortex parallel electroencephalographic differences among mouse strains. J Neurosci 28:7193-7201.